# Durable Oral Biofilm Resistance of 3D-Printed Dental Base Polymers Containing Zwitterionic Materials

**DOI:** 10.3390/ijms22010417

**Published:** 2021-01-03

**Authors:** Jae-Sung Kwon, Ji-Yeong Kim, Utkarsh Mangal, Ji-Young Seo, Myung-Jin Lee, Jie Jin, Jae-Hun Yu, Sung-Hwan Choi

**Affiliations:** 1Department and Research Institute of Dental Biomaterials and Bioengineering, Yonsei University College of Dentistry, Seoul 03722, Korea; jkwon@yuhs.ac; 2BK21 FOUR Project, Yonsei University College of Dentistry, Seoul 03722, Korea; katekim826@yuhs.ac (J.-Y.K.); hun718@yuhs.ac (J.-H.Y.); 3Department of Orthodontics, Institute of Craniofacial Deformity, Yonsei University College of Dentistry, Seoul 03722, Korea; utkmangal@yuhs.ac (U.M.); jyseo13@yuhs.ac (J.-Y.S.); kimj0515@yuhs.ac (J.J.); 4Department of Dental Hygiene, Division of Health Science, Baekseok University, Cheonan 31065, Korea; dh.mjlee@gmail.com

**Keywords:** dentistry, dental base resin, 3D printing, poly(methyl methacralyate), zwitterion, oral salivary biofilm, durability

## Abstract

Poly(methyl methacralyate) (PMMA) has long been used in dentistry as a base polymer for dentures, and it is recently being used for the 3D printing of dental materials. Despite its many advantages, its susceptibility to microbial colonization remains to be overcome. In this study, the interface between 3D-printed PMMA specimens and oral salivary biofilm was studied following the addition of zwitterionic materials, 2-methacryloyloxyethyl phosphorylcholine (MPC) or sulfobetaine methacrylate (SB). A significant reduction in bacterial and biofilm adhesions was observed following the addition of MPC or SB, owing to their protein-repellent properties, and there were no significant differences between the two test materials. Although the mechanical properties of the tested materials were degraded, the statistical value of the reduction was minimal and all the properties fulfilled the requirements set by the International Standard, ISO 20795-2. Additionally, both the test materials maintained their resistance to biofilm when subjected to hydrothermal fatigue, with no further deterioration of the mechanical properties. Thus, novel 3D-printable PMMA incorporated with MPC or SB shows durable oral salivary biofilm resistance with maintenance of the physical and mechanical properties.

## 1. Introduction

Poly(methyl methacralyate) (PMMA) has long been used in dentistry for the production of base polymers in dentures, as well as other removable dental devices such as orthodontic retainers or occlusal splints used for temporomandibular joint therapy [1], due to their advantages such as cost effectiveness, reasonable mechanical properties, stability in oral environments and excellent esthetic characteristics [2].

In recent years, major advancements in the application of three-dimensional (3D) printing technology in dentistry has led to the production of patient-specific prostheses in a cost-effective and time-saving manner [3,4]. 3D-printing materials that are commonly used for dental restoration include the light-curing polymer resin which produce 3D-printed products by converting liquid matters to a solid under the action of light such as ultraviolet or visible light with technology such as stereolithography (SLA) [5]. PMMA has also been commonly used as an SLA-printable dental material, owing to its favorable properties such as light-curability, flexibility, formability, biocompatibility, and cost-effectiveness [1,6].

Despite the aforementioned advantages, there are still certain technical limitations of using PMMA as a dental base polymer. One major limitation is its great susceptibility to microbial colonization owing to microbial adhesion followed by biofilm formation on dentures, resulting in local infection of the oral cavity and systemic infections such as aspirational pneumonitis [7,8]. Various methods have been considered to overcome this drawback, including the use of antibiotics or bioactive fillers, although it is well established that antibiotics do not work with established mature biofilms, and the effects of these bioactive agents are transient and some degrade the mechanical properties of the device [9,10,11].

Zwitterionic materials are a family of materials with both cationic and anionic moieties; they are characterized by high dipole moments and consist of highly charged groups, but have an overall neutral charge, which enables their application as antifouling materials as they can form a hydration shell via electrostatic interactions, which are much stronger than hydrogen bonds, resulting in denser and more tightly adsorbed water [12]. 2-methacryloyloxyethyl phosphorylcholine (MPC) and sulfobetaine methacrylate (SB) are widely used zwitterionic materials owing to their biocompatibility and their ease of production and use [13,14].

Previously, MPC and SB have been used in dental materials such as resin-based composites, fluoride varnishes, and calcium and silicate cements [15,16,17]. Studies have shown the protein-repellent and antibacterial effects of these dental materials without the deterioration of other properties relevant for clinical use, and their favorable features have also been shown to be durable. Such protein-repellent and consequent anti-fouling effects of two zwitterionic materials are thought to be linked to the hydrophilicity and nature of charge of the two materials [18,19]. In the state of hydrated polymer, there is an abundance of free water but no bound water in such zwitterionic materials [18]. The presence of bound water would cause protein adsorption [18] whereas the large amount of free water around the functional group of the molecule would detach proteins, effectively causing protein-repelling properties [18]. In this study, we successfully produced a 3D-printable PMMA dental base material incorporated with MPC or SB. The purpose of this in vitro study was to evaluate the effects of MPC or SB on the interface between a PMMA-based dental device and an oral salivary biofilm. The deterioration of any clinically relevant features of the 3D-printed PMMA specimens was also studied in addition to the durability of the feature related to each zwitterionic material. The null hypotheses were that there will be no significant differences between PMMA with or without MPC/SB in terms of the (i) physical and mechanical properties, (ii) oral salivary biofilm resistance, and (iii) these two properties following hydrothermal fatigue.

## 2. Results

### 2.1. Physical and Mechanical Properties of 3D-Printed PMMA

The addition of either zwitterion into PMMA resulted in a significant reduction in the contact angle compared to that of the control (*p* < 0.001) (Figure 1A). Following the thermocycling process, there was no significant difference between SB-incorporated PMMA and the control. However, MPC-incorporated PMMA maintained a significantly lower contact angle than the control (*p* < 0.05) (Figure 1B).

Both the flexural strength and the elastic modulus of PMMA decreased following the addition of MPC or SB (*p* < 0.001) (Figure 1C,E). However, the difference between the mean values was small (~15 MPa for strength and ~350 MPa for modulus) compared to those of the control. Remarkably, the flexural strength and elastic modulus of all the samples exceeded the minimum requirements (50 MPa for strength and 1500 MPa for modulus) set by ISO 20795-2 [16,20]. A similar trend was maintained following the thermocycling process, and there were no significant differences between the elastic moduli of the control and SB-incorporated PMMA (Figure 1D,F).

Finally, the addition of either zwitterion resulted in a statistically significant reduction in the Vickers hardness (VHN) (*p* < 0.001) (Figure 1G). The approximate difference is 2 VHN. Similar results were obtained after thermocycling (Figure 1H).

### 2.2. Protein Adsorption on 3D-Printed PMMA

Significant reduction in the BSA adsorption on both test groups was evident compared to that of the control (*p* < 0.001), and there were no significant differences between MPC- and SB-containing PMMA specimens (*p* > 0.05) (Figure 2A). The same trend was observed with proteins from the BHI (brain heart infusion) medium (Figure 2B).

### 2.3. Bacterial Adhesion on 3D-Printed PMMA

The live/dead images of the samples revealed a larger number of green live bacteria on the control than on the MPC- or SB-incorporated PMMA for all four different bacteria (Figure 3A). In terms of quantitative analyses, it was evident that both the samples printed from MPC- and SB-incorporated PMMA had a significantly lower number of attached bacteria than the control (*p* < 0.001) (Figure 3B). Furthermore, there were no statistical differences in the bacterial attachment between the two test samples with the zwitterion.

The results were confirmed with quantitative analyses by colony forming unit (CFU) counts of *Streptococcus mutans* before and after exposure to thermocycling (Figure 4). The results confirmed the above findings, as the results indicated that there were significant reductions in CFU counts for both MPC- and SB-incorporated PMMA, while there were no statistical differences between the two test samples with the zwitterion. Furthermore, such reductions in CFU counts for both MPC- and SB-incorporated PMMA were maintained even after exposure to thermocycling.

### 2.4. Human Salivary Oral Biofilm on 3D-Printed PMMA

The graphical live/dead images indicate an abundant biofilm formed on the control samples, whereas green-stained bacteria were sparse on PMMA containing MPC or SB (Figure 5A). Similar trends were noted following thermocycling, although there seems to be an increase in the number of green-stained bacteria, which indicates an increase in biofilm attachment on the MPC- and SB-containing PMMA after thermocycling.

Further, both the biofilm thickness and the biomass were significantly reduced following the addition of MPC or SB (*p* < 0.01) (Figure 5B). Although the statistical significance disappeared between the control and MPC-incorporated PMMA following thermocycling, the control samples had a ~1.4-times thicker biofilm than the MPC-based samples in terms of average values. Otherwise, all the test samples showed significantly reduced biofilm thickness and biomass, even after thermocycling (*p* < 0.05) (Figure 5B).

### 2.5. Dimensional Accuracy of 3D-Printed PMMA

The green coloration in the superimposed optical scans (Figure 6) indicates tolerable dimensional changes (the RMS (root-mean-square) value was set at ±0.05 µm) from those of the scanned digital file, while an RMS deviation of 0.2 µm is indicated by red or blue colorations in accordance to positive or negative RMS values, respectively. Notably, the produced samples are green in most areas although some red or blue coloration is evident for all samples. The RMS values are calculated to be 110.00 ± 13.50, 101.83 ± 17.37, and 104.50 ± 6.64 µm for PMMA, PMMA with MPC, and PMMA with SB, respectively. There were no statistical differences between the three printed samples.

## 3. Discussion

Numerous attempts have been made to impart antibacterial and biofouling properties to PMMA-based dental devices; these include the incorporation of silver nanoparticles/ions [5], titanium dioxide [21], and other nanomaterials [22]. However, the durability of the added agents has often been questioned, and the fabricated materials have shown limited mechanical strength for the long-term application in functional dental parts.

Hence, this study considered the development of a novel 3D-printable PMMA-based formulation that would resist the growth of human salivary biofilms, while maintaining the original advantageous physical and mechanical properties.

The first null hypothesis of this study was that there would be no significant differences between the physical and mechanical properties of PMMA with and without MPC/SB. This hypothesis was partially validated. The clinical success and cost-effectiveness of a denture base or orthodontic retainer are closely related to the physical and mechanical properties of the device produced from PMMA [23,24,25]. The results revealed a significant decline in the flexural strength, elastic modulus, and Vickers hardness following the addition of MPC or SB (Figure 1). Previous studies have indicated that some physical and mechanical properties deteriorate with the addition of a high dose of zwitterions [26]. These have been attributed to the gelation of the materials at higher zwitterionic concentrations, which consequently interferes with graft polymerization between the zwitterions and PMMA [26,27]. The concentration of MPC and SB in this study was based on previous studies, which examined different concentrations of MPC and SB. In those studies, more than 5 wt% of the zwitterion resulted in the deterioration of the mechanical properties, while less than 3 wt% led to less effective protein-repellent properties and biofilm resistance [15,16,17,28]. Despite a small decline, the mechanical properties obtained in this study are adequate in terms of the minimum required values stated in International Standard ISO 20795-2 [20].

The second null hypothesis was that there would be no significant differences between the oral salivary biofilm resistance of PMMA with and without MPC/SB. This hypothesis was rejected because bacterial adhesion on MPC- or SB-containing PMMA was clearly inhibited with all the early colonizers for oral biofilms (Figure 3 and Figure 4). The mechanism is related to the unique structure of either zwitterion, which allows a large amount of free water to be present around the functional group, whereas there would be no bound water in the hydrated zwitterions [14]. As free water would repel protein adsorption, the incorporation of a zwitterion in PMMA results in protein-repellent properties [29]. This leads to resistance to bacterial adhesion, because the initial step in this process is the adsorption of salivary proteins as a salivary pellicle that can mediate bacterial attachment and biofilm formation [30]. Indeed, significantly reduced adsorption of BSA and proteins in BHI medium were also identified in this study (Figure 2).

The adhesion of early colonizers by the salivary pellicle, which is the initial step of biofilm formation, was inhibited by the zwitterions in PMMA, and the results of the biofilm studies were predictable. This study was conducted to consider attachment of biofilm on the surface of test and control materials, with indications of biofouling by biomass and thickness. Initial analyses on such oral salivary biofilms have been considered in previous studies that used the same sample of oral salivary biofilm as in the current study [17]. The results revealed a significant reduction in the biomass and thickness in the oral saliva-derived biofilm on the test groups compared to the control (Figure 5). Previous studies have reported that adhesion and adsorption of saliva-derived protein is an initial step in biological interactions between oral biofilms and materials present in human oral cavities [31]. Zwitterionic materials with polar groups in the side chain have similar structures to lipid bilayer structures of cell membranes [32], where the molecule consists of a hydrophilic head (attracted to water) and a hydrophobic tail (repelled by water) [32]. Hence, when such molecules are exposed to oral environments, the interaction between water molecules and zwitterionic molecules leads to a large amount of free water around the zwitterionic molecules which detach and repel proteins in human saliva [18,33].

Interactions between dental plaque and dental material surfaces would have similar results. Dental plaque are aggregates of microorganisms, which are formed due to the attachment of bacteria in the oral environment following initial formation of a salivary pellicle layer on the tooth or material surfaces [34]. Salivary pellicles are formed by selective adsorption of salivary proteins, and hence adhesion and formation of dental plaque would be prevented by prevention of the initial step of salivary pellicle formation [34,35].

The ability of zwitterion-incorporated PMMA to inhibit bacterial adhesion and consequently reduce biofilm adhesion was maintained even after undergoing hydrothermal fatigue by a thermocycling process (Figure 5). This result validates the last hypothesis of this study, i.e., that there will be no significant differences between PMMA with and without MPC/SB in terms of the physical/mechanical properties and oral salivary biofilm resistance following hydrothermal fatigue. The thermocycling process simulates the intraoral physical conditions, during which hydrolytic aging of some of the materials could occur, resulting in plasticization and deterioration of the mechanical and biological properties owing to the damage of polymer chains (e.g., cleavage of ester bonds), as observed in previous studies [36,37]. Despite undergoing such fatigue, minimal or no changes were observed in the materials investigated in this study. It has been previously demonstrated that the incorporation of zwitterions in resin-like materials results in the copolymerization and immobilization of zwitterions [38], resulting in a stable form of the material. The results presented here are in agreement with those of previous studies, wherein no significant changes were observed with zwitterion-containing polymers in terms of both the protein-repellent and the physical/mechanical properties [17,28].

Finally, the added MPC or SB had no influence on the dimensional accuracy of the 3D-printed products. For the clinical success of the patient-specific production of PMMA-based dental base resins, dimensional accuracy is an important requirement for 3D-printable PMMA [24,39]. Here, a clinically relevant model was produced from PMMA with or without the zwitterion. Although all the samples showed some variations compared to the standard, there were no statistical differences between the control and test materials, indicating the possibility of using MPC- or SB-incorporated PMMA as a 3D-printable dental resin.

In this study, ex vivo experiments were not conducted using a complicated model in the clinical oral environment, such as under the salivary flow and the presence of food debris, and these should be considered in future in vivo or clinical studies. Furthermore, future studies are planned to consider how such MPC- or SB-containing PMMA materials influence the oral microbiota related to salivary biofilm. Despite this limitation, this study indicates that the addition of MPC or SB into PMMA results in durable oral salivary biofilm resistance, with the maintenance of physical and mechanical properties.

## 4. Materials and Methods

### 4.1. Preparation of Zwitterion-Incorporated 3D-Printable PMMA

Two zwitterionic materials, viz., MPC and SB (Sigma-Aldrich, St. Louis, MO, USA) were used in this study. Each zwitterionic powder (3 wt%) was homogeneously mixed with distilled water using a speed mixer at 3500 rpm for 2 min and then mixed with PMMA for 3D printing (NextDent Ortho Rigid, 3D Systems, NextDent B.V., Soesterberg, The Netherlands). The PMMA specimens were 3D-printed using a digital light processing 3D printer (NextDent 5100, 3D Systems, NextDent B.V.). Thereafter, the 3D-printed blocks were detached from the platform and washed with isopropyl alcohol to remove excess resin monomers. The samples were post-processed for 10 min using a UV oven (NextDent LC-3DPrint Box, 3D Systems, NextDent B.V.). After curing, the samples were sequentially polished to 800, 1500, and 2000 grit.

### 4.2. Physical and Mechanical Properties of 3D-Printed PMMA

To evaluate the wettability of each sample, disc-shaped samples (n = 7 per group) were printed (diameter, 10 mm; thickness, 2 mm). A 5 µL droplet of distilled water was dropped on each sample and the contact angle was measured after 10 s using a video contact angle goniometer (SmartDrop, Femtobiomed Inc., Gyeonggi-do, Korea).

The flexural strength and elastic modulus were evaluated according to methods adopted from International Standard ISO 20795-2 Dentistry–Base polymers–Part 2: Orthodontic base polymers [16]. Samples (n = 10 per group) with dimensions of 64.0 × 10.0 × 3.3 mm were printed. Before the mechanical test, all the samples were stored at 37 °C in distilled water for 48 h. A universal testing machine (Model 3366, Instron, Norwood, MA, USA) was used for three-point flexure testing. The span length and crosshead speed were 50 mm and 5 mm/min, respectively.

The Vickers hardness was measured using a hardness tester (DMH-2, Matsuzawa Seiki Co., Ltd., Tokyo, Japan) under an established load of 300 gf (2.94 N) for 30 s. The average value for each sample (n = 5 per group) was calculated from measurements at three sites.

All the experiments were repeated after subjecting each specimen to hydrothermal fatigue. In this study, hydrothermal fatigue was considered to stimulate the intraoral physical environment in order to evaluate the durability of each material. The method was adopted from previous reports [17,18]. Briefly, thermocycling was conducted between 5 and 55 °C with a dwell time of 45 s for 850 cycles, corresponding to 1 month.

### 4.3. Protein Adsorption on 3D-Printed PMMA

Protein adsorption was tested using a previously established method [13,15]. Each material was molded into disc-shaped samples (diameter, 15 mm; thickness, 2 mm) and the sample discs were immersed in fresh phosphate-buffered saline (PBS; Gibco, Grand Island, NE, USA) for 1 h at room temperature and then were immediately immersed in a protein solution (2 mg/mL in PBS; 100 μL) of either bovine serum albumin (BSA; Pierce Biotechnology, Rockford, IL, USA) or brain heart infusion (BHI; Difco, Sparks, NV, USA) broth. After incubation at 37 °C for 1 h, the samples were gently rinsed twice with fresh PBS. After 4 h of incubation under sterile humid conditions at 37 °C, any unadhered protein was removed by washing twice with PBS. The amount of protein adhered to the samples was determined using 200 μL of micro-bicinchoninic acid (Micro BCA^TM^ Protein Assay Kit; Thermo Fisher Scientific Inc, MA, USA) followed by incubation at 37 °C for 30 min. The amount of the proteins adsorbed on the surfaces was quantified by measuring the absorbance at 562 nm using a micro-plate reader (Epoch, BioTek Instruments, VT, USA).

### 4.4. Bacterial Adhesion on 3D-Printed PMMA

Bacterial analyses were carried out using four different bacterial species, *Streptococcus mutans* (ATCC 25175), *Staphylococcus aureus* (ATCC 25923), *Klebsiella oxytoca* (KCOM 1569), and *Klebsiella pneumoniae* (KCOM 2770), which are known to be early colonizers of oral biofilms and therefore are the key pathogens related to biofilm adhesion on the dental base polymer, PMMA [19].

The samples for bacterial experiments were sterilized with ethylene oxide gas. All bacteria were cultured in BHI broth under aerobic incubation at 37 °C for 18 h or longer. Then, 1 mL (1 × 108 cells/mL) of the cultured bacterial suspension was dispensed on the sample and cultured again at 37 °C for 24 h. The samples were washed twice by gently shaking in PBS to remove any unattached bacteria.

The viability of the adherent bacteria was examined by staining with a live/dead bacterial viability kit (Molecular Probes, Eugene, OR, USA), according to the manufacturer’s protocols. After 20 min of storage at room temperature in the dark, the stained samples were observed under a confocal laser microscope (LSM880; Carl Zeiss, Thornwood, NY, USA).

Bacterial colony forming units (CFU) of *S. mutans* were evaluated twice before and after thermocycling. The sample was immersed in a bacterial suspension of the concentration 1 × 108 cells/mL at 37 °C for 24 h. In the same way as above, after gently rinsing in PBS, it was soaked in 1 mL of PBS and sonicated (SH-2100; Saehan Ultrasonic, Seoul, Korea) for 5 min. The bacterial suspension separated from the sample by sonication was serially diluted and spread onto BHI agar plate. After incubation at 37 °C for 24 h, the total numbers of colonies were counted.

### 4.5. Human Salivary Oral Biofilm on 3D-Printed PMMA

Studies using human saliva-derived biofilms were based on a previous report [15] and adopted from the guidelines related to the interface of materials and microbiology [20]. The biofilm model was prepared by mixing equal proportions of the human saliva of six adults who did not have any periodontal disease or active caries and had not taken antibiotics within the past 3 months; they voluntarily provided saliva following approval by the Institutional Review Board (IRB No. 2-2019-0049). The biofilm model was cultured in a supplemented McBain medium at 37 °C. From the cultured medium, 1.5 mL of the bacterial suspension was dropped onto each sample. After 8, 16, and 24 h of incubation, respectively, 1.5 mL of the McBain medium was additionally placed on the sample and the biofilms were allowed to grow for 48 h. The biofilms were then stained with a live/dead bacterial viability kit and observed as described above. The thickness and biomass of the biofilm were assessed using the COMSTAT plug-in (Technical University of Denmark, Copenhagen, Denmark) along with ImageJ software (NIH). The entire process was repeated following the hydrothermal fatigue of the samples by the same methods described above for evaluating the physical and mechanical properties.

### 4.6. Dimensional Accuracy of 3D-Printed PMMA

The dimensional accuracy of 3D-printed PMMA with or without the zwitterion was evaluated according to a previously reported method [21]. The image acquired from the standard orthodontic study model using 3Shape E3 Scanner (3Shape, Copenhagen, Denmark) was used to 3D print denture-base models using PMMA with or without MPC/SB. Dimensional differences between the original scanned data and three different groups were compared by the best-fit superimposition method using a 3D morphometric program (Geomagic Control X, 3D Systems, SC, USA). The software was used to calculate root-mean-square (RMS) values, which indicate the trueness values of each produced sample against the original scanned data.

### 4.7. Statistical Analysis

The statistical analyses were performed using data from at least three independent experiments using statistical software (IBM SPSS, version 23.0, IBM Korea Inc., Seoul, Korea). The results were analyzed by one-way analysis of variance (ANOVA) and post-hoc Tukey’s test. *p* < 0.05 was considered to be statistically significant.

## Figures and Tables

**Figure 1 ijms-22-00417-f001:**
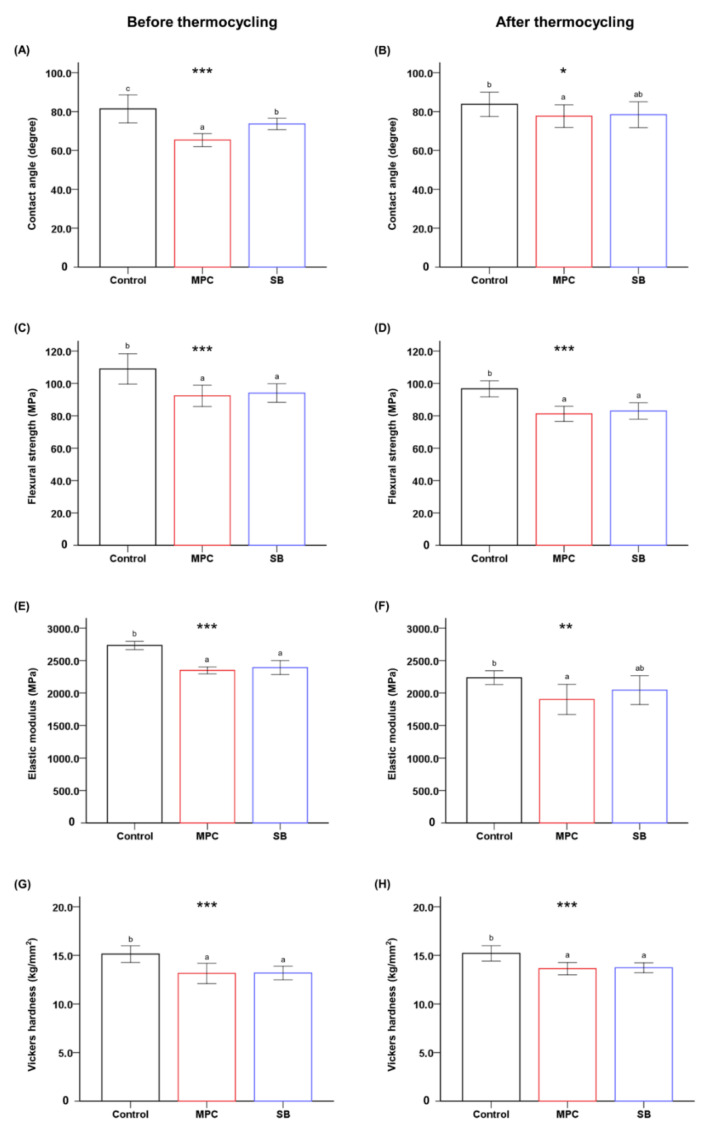
Physical and mechanical properties in terms of the contact angle (**A**,**B**), flexural strength (**C**,**D**), elastic modulus (**E**,**F**), and Vickers hardness (**G**,**H**) of poly(methyl methacralyate) (PMMA) before (Control) and after the addition of 3 wt% 2-methacryloyloxyethyl phosphorylcholine (MPC) or sulfobetaine methacrylate (SB). Results obtained before (left) and after thermocycling (right) are shown. Different lowercase letters above the bars indicate significant differences by post-hoc Tukey’s test. *** *p* < 0.001, ** *p* < 0.01, * *p* < 0.05 for comparisons between PMMA with and without the zwitterion analyzed by one-way analysis of variance (ANOVA).

**Figure 2 ijms-22-00417-f002:**
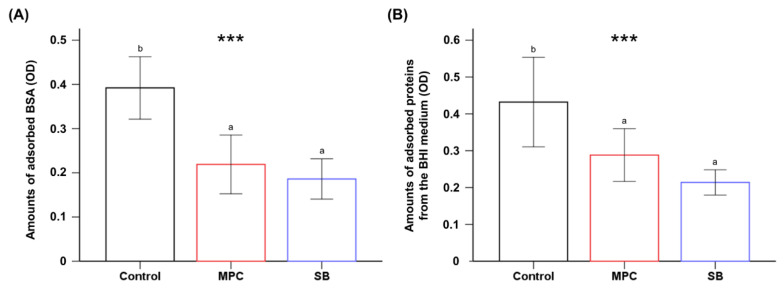
Comparison of the optical density (OD) of the adsorbed bovine serum albumin (BSA) (**A**) and protein adsorbed from the brain heart infusion (BHI) medium (**B**) of poly(methyl methacralyate) PMMA samples before (Control) and after the addition of 2-methacryloyloxyethyl phosphorylcholine (MPC) or sulfobetaine methacrylate (SB). Different lowercase letters above bars indicate significant differences by post-hoc Tukey’s test. *** *p* < 0.001 for comparisons between PMMA with and without the zwitterion analyzed by one-way analysis of variance (ANOVA).

**Figure 3 ijms-22-00417-f003:**
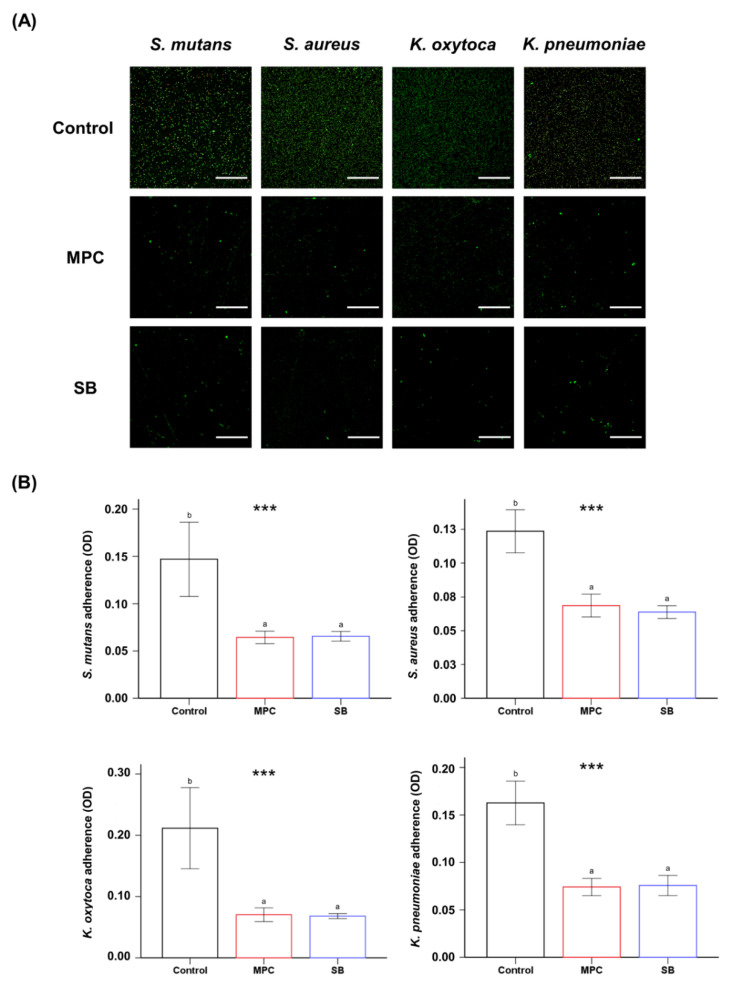
(**A**) Live and dead images of four different bacteria (*S. mutans*, *S. aureus*, *K. oxytoca* and *K. pneumoniae*) attached to poly(methyl methacralyate) (PMMA) before (Control) and after the addition of 2-methacryloyloxyethyl phosphorylcholine (MPC) or sulfobetaine methacrylate (SB). Green coloration indicates live bacteria (scale bar, 100 µm). (**B**) Optical density (OD) readings derived from bacteria attached on the surfaces of the Control, MPC-incorporated PMMA, and SB-incorporated PMMA. Different lowercase letters above the bars indicate significant differences by post-hoc Tukey’s test. *** *p* < 0.001 for comparisons between PMMA with and without the zwitterion analyzed by one-way analysis of variance (ANOVA).

**Figure 4 ijms-22-00417-f004:**
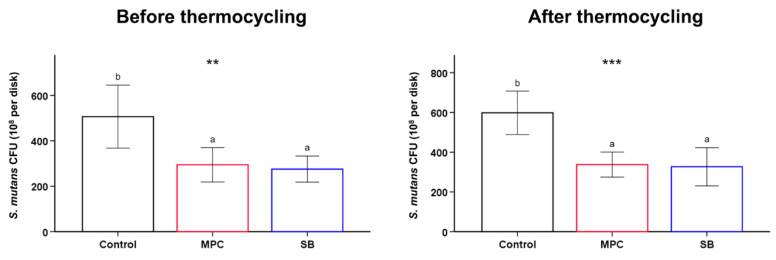
Colony forming unit (CFU) counts of *S. mutans* attached to poly(methyl methacralyate) (PMMA) (Control) and PMMA incorporated with 2-methacryloyloxyethyl phosphorylcholine (MPC) or sulfobetaine methacrylate (SB), before and after thermocycling. Different lowercase letters above the bars indicate significant differences by post-hoc Tukey’s test. *** *p* < 0.001, ** *p* < 0.01 for comparisons between PMMA with and without the zwitterion analyzed by one-way analysis of variance (ANOVA).

**Figure 5 ijms-22-00417-f005:**
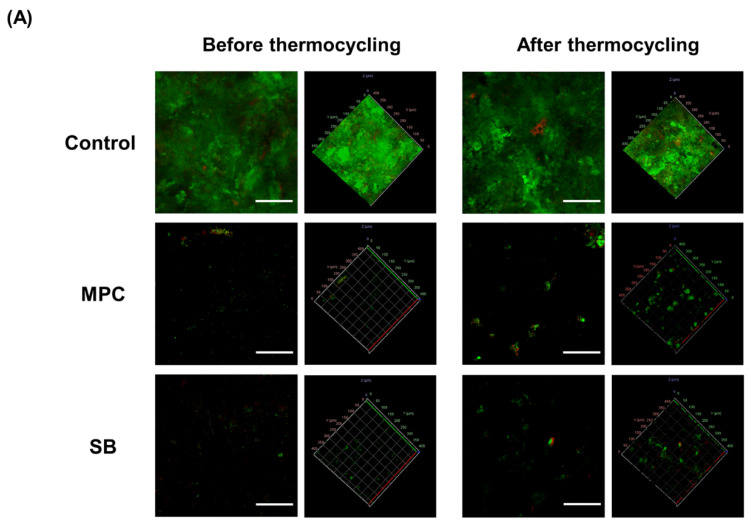
(**A**) Live/dead staining images of biofilms attached to the surfaces of poly(methyl methacralyate) (PMMA) before (Control) and after the addition of 2-methacryloyloxyethyl phosphorylcholine (MPC) or sulfobetaine methacrylate (SB). Scale bar is 100 µm. (**B**) Quantitative analysis of the thickness and biomass of the biofilms. Different lowercase letters above the bars indicate significant differences by post-hoc Tukey’s test. ** *p* < 0.01, * *p* < 0.05 for comparisons between PMMA with and without the zwitterion analyzed by one-way analysis of variance (ANOVA). Results obtained before thermocycling (left) and after thermocycling (right) are shown.

**Figure 6 ijms-22-00417-f006:**
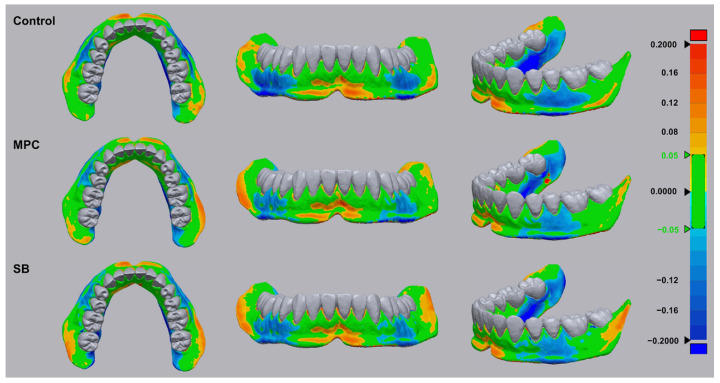
Superimposed optically scanned surfaces of 3D-printed dental base-like models constructed using poly(methyl methacralyate) before (Control) and after the addition of 2-methacryloyloxyethyl phosphorylcholine (MPC) or sulfobetaine methacrylate (SB). The calculated root-mean-square (RMS) values within tolerable dimensional changes with respect to those of the scanned digital file are indicated by green coloration (RMS value is set at ±0.05 µm). Red and blue indicate +0.2 µm and −0.2 µm RMS deviations from the scanned digital file, respectively.

## Data Availability

All relevant data are within the manuscript.

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
