# Peer review of "Durable Oral Biofilm Resistance of 3D-Printed Dental Base Polymers Containing Zwitterionic Materials"

_ijms, 2021, doi:10.3390/ijms22010417_

Round 1
Reviewer 1 Report
It is an article of interest, correctly carried out, and its publication will be scheduled.
Author Response
Comment: It is an article of interest, correctly carried out, and its publication will be scheduled.
Reponse: Thank you very much for your encouraging comments. We now have updated manuscript further and hope this would be adequate for publications.
Reviewer 2 Report
introduction is poor. Materials should be better introduced and compared with other ones. Interface with cells and tissues are not well described. Biological aspects need to be reported.
Author Response
Comment: introduction is poor. Materials should be better introduced and compared with other ones. Interface with cells and tissues are not well described. Biological aspects need to be reported.
Reponse: Thank you very much for your comment and sorry for limited information. First of all, we now have updated Introduction. Information on materials considered in this study in terms of their background have been added. Interface with cells/tissues in terms of oral environment have now been considered with additional references in Discussion.
Reviewer 3 Report
In this study, the interface between 3D-printed PMMA specimens and oral salivary biofilm was studied following the addition of a zwitterionic material, 2-methacryloyloxyethyl phosphorylcholine (MPC) or sulfobetaine methacrylate (SB). Authors claim that significant reduction in bacterial and biofilm adhesions was observed in the samples additioned with MPC or SB as compared to PMMA only and there were no significant differences between these two materials.
This is an interesting work and the manuscript is well written. The experiments performed were mostly adequate and support the Authors’ claim. However, there are a few major and minor points that have to be addressed, in my opinion:
Major points:
1) A few live bacteria can initiate a biofilm formation. Therefore, it would be important to assess CFU counts of the attached bacteria to the different surfaces studied (in particular, post thermocycling). The confocal microscope images are not sufficient to quantify the number of bacteria as it depends on the area examined. The optical density measurement does not distinguish between dead and live bacteria.
2) For human salivary oral biofilm experiments the initial bacteria count (CFU/piece if identical or CFU/mm2) and the CFU of biofilm associated bacteria (after 48h) should be assessed.
3) With which statistical method the statistical analyses are performed ? This information is missing. It should be inserted to Materials and Methods section and to each figure legend where a statistical analysis result is reported.
Minor point:
Figures 1,2,3,4: The indication of statistical results on the bars with the letters is confusing (i.e. 3 asterisks on MPC bar indicate significance compared to control is not clear from the figure, what about SB?). Please clearly indicate the statistical significance of the conditions compared (e.g. with an addition of a line between compared conditions etc.).
Author Response
Major points:
Comments 1): A few live bacteria can initiate a biofilm formation. Therefore, it would be important to assess CFU counts of the attached bacteria to the different surfaces studied (in particular, post thermocycling). The confocal microscope images are not sufficient to quantify the number of bacteria as it depends on the area examined. The optical density measurement does not distinguish between dead and live bacteria.
Response 1): Thank you very much for your encouraging comments and also some of major and minor points for revision. We are sorry about lack of information. We now have conducted additional experiments to consider CFU counts of S. Mutans before and after thermocycling. The results are included as new Figure, Figure 4. Also materials and method for the results are now included and updated.
Comments 2): For human salivary oral biofilm experiments the initial bacteria count (CFU/piece if identical or CFU/mm2) and the CFU of biofilm associated bacteria (after 48h) should be assessed.
Reponse 2): Thank you for your comment. Our initial analyses on biofilm has been already included in other studies that used same sample oral saliva and such information is now added to Discussion. Also, our focus with human salivary biofilm model was to consider biothickness and biomass related to biofouling and not with killing of bacteria. Information with individual bacteria were considered in previous figure while further studies with influence in oral microbiota are currently considered in other studies. All these are now included in Discussion to provide more clear rationale. Thank you for your comments.
Comment 3): With which statistical method the statistical analyses are performed ? This information is missing. It should be inserted to Materials and Methods section and to each figure legend where a statistical analysis result is reported.
Response 3): Sorry for such mistake. We now have included statistical anlyses, which were carried out by ANOVA with post-hoc Tukey's test. This is also linked to minor comment below.
Minor point:
Comments: Figures 1,2,3,4: The indication of statistical results on the bars with the letters is confusing (i.e. 3 asterisks on MPC bar indicate significance compared to control is not clear from the figure, what about SB?). Please clearly indicate the statistical significance of the conditions compared (e.g. with an addition of a line between compared conditions etc.).
Response: Thank you and sorry for the confusion. Bar with letters are from post-hoc analyses (Tukey's) while asterisks are related to ANOVA. This information is now included in Figure legends and Materials and Method. Due to multiple groups we tried line between compared condition but it was not tidy/clear. We hope that improvement is clear.
Round 2
Reviewer 3 Report
The manuscript is acceptable in its present form